# Coastal Monitoring Using Unmanned Aerial Vehicles (UAVs) for the Management of the Spanish Mediterranean Coast: The Case of Almenara-Sagunto

**DOI:** 10.3390/ijerph19095457

**Published:** 2022-04-29

**Authors:** Vicent Esteban Chapapría, José Serra Peris, José A. González-Escrivá

**Affiliations:** Ports & Coastal Engineering, Universitat Politècnica Valencia, 46021 Valencia, Spain; jserra@upv.es (J.S.P.); jgonzale@upv.es (J.A.G.-E.)

**Keywords:** population, monitoring, coastal erosion, integrated coastal zone management, Unmanned Aerial Vehicles (UAVs)

## Abstract

The concentration of the world’s population in coastal areas means an increase in pressure on the environment and coastal ecosystems. The impacts of climate change affect natural biophysical and ecological systems and human health. Research has been developed to create coastal monitoring with Unmanned Aerial Vehicles (UAVs) that allow data to be obtained and methodologies that integrate computer vision algorithms for 3D and image processing techniques for analysis, combined with maritime information. The Valencian oval is located on the Spanish Mediterranean coast and registers significant coastal erosion. It is a densely populated area, with high economic relevance and tourist activity. The main goals of the developed research in this coastal area include creating a methodology of data collection that identifies environmental indicators significant to community health and uses in the coastal areas, to test progression of interventions and to assess coastal erosion detection and monitoring. The final objective is to aid in decision-making and coastal management. Sediment characterization was obtained, and continuous maritime information was collected. The dynamic evolution of coastal areas was researched by using UAVs on the Spanish Mediterranean coast. This technique is suitable for measuring medium to small coastal changes. Flight planning was carried out using the grid mode and adapted to areas in order to obtain a homogeneous pixel size and precision. This monitoring program takes advantage of technological development with very low economic costs and is a good tool for making decisions that must be based on scientific information. With the monitoring work, an annual erosion between 12 and 6 m was detected. The monitoring program has evidenced the shoreline trend as a result of the impact of rigid structures, mainly ports and groins, in promoting down-drift erosion processes in the area.

## 1. Introduction

The impacts of climate change affect natural biophysical and ecological systems and human health [1,2,3,4]. Coastal erosion is affecting the environment and economic activities globally [5]. A wide spectrum of different processes is involved [6] at different temporal and spatial scales, including hydrodynamic and other processes. The most at-risk populations are those on the coast, due to their large size and high exposure to rising sea levels, storm surges, coastal erosion and river flooding, which is increased by human-induced pressures on coastal areas [7,8].

Since the 1950s, coastal areas have steadily increased in population [9]. This is a generalized process throughout the world. In Spain, one in three people live in a town on the coast [10]. The concentration of the world’s population in coastal areas means an increase in pressure on the environment and coastal ecosystems. Projections indicate that by 2050, 70% of the world’s population will live in urban areas and agglomerations [11]. This is reflected in the characteristics of concentration and rapid and sustained growth. In turn, however, there are two other characteristics to consider: dispersion of the population and conurbation of two or more cities. Therefore, the study of coastal urban development on a global scale is of great interest for the integrated management of coastal areas.

Research on the evolution of cities and coastal agglomerations between 1945 and 2012 established that the number and size of cities, and the populations living in them, are continuously increasing. As a result, the concept of city and coastal agglomeration (CCA) is considered as referring to those with more than 100,000 inhabitants located less than 100 km from the coastline. In turn, however, there are two other characteristics to consider: dispersion of the population and conurbation of two or more cities. Mechanisms to determine the relationship between various ecological, geographic and socioeconomic factors and urban development on the coast have been analyzed [9]. An investigation was carried out using UN population census data with the use of Google Earth, which allowed for the observation of cities and their development, as well as associated infrastructure, especially ports and roads, and their relationship with the area. There are currently more than 2100 CCAs with more than 100,000 inhabitants in the world, where almost 1500 million people live [9]. In addition to this population, there are also other smaller population centers and those of coastal rural areas. There are areas throughout all Mediterranean countries where this is so. Many of these coastal areas are tourist destinations and tourism economic impact is relevant. CCAs refer to spaces that relate directly, fluidly, intensely and in a double direction to the terrestrial social environment and marine coastal ecosystems. Consequently, there is convergence between the integrated coastal zone management (ICZM) and urban management. As a result, they are of extraordinary importance for the integrated management of coastal areas [12,13,14,15,16].

Coastal erosion is a process that degrades coastal profiles and mainly occurs due to natural factors (e.g., related to climate change) and overcrowding (e.g., urbanization and massive tourism) [17]. While the first cause can be considered as a slow process, the growing presence of humans is leading to the rapid aging of coasts. Mediterranean Sea authorities are focusing on the necessity for a systematic and comprehensive approach to the management of littoral areas. Coastal areas are at risk due to rising sea levels associated with climate change. Therefore, the stability of these areas is becoming increasingly important, and sedimentary materials play an increasingly important role in determining the amount of damage from flooding and erosion [18]. The fundamentals of sedimentary transport have been studied [19,20,21,22]. Some research on used management approaches in the studied area demonstrate the inefficiency of hard measures to stop erosion and their negative impact on environmental quality [23]. Scale and resolution of the images used in some investigations which provide very good information on the evolution of the coast is large [24]. Due to varying processes and their different scales, data acquisition and the analysis of coastal evolution and its impacts have not been periodically systematized in coastal management practices [25]. Coastal evolution and erosion have been monitored in a variety of ways [12,26,27,28]. It has been promoted at regional and municipal levels [7,29,30,31,32]. It has been executed on dunes, estuaries, port environments, cliffs, etc., and coastal observatories have been created in recent years [25,29,33,34,35]. Research must explain and properly combine time and space scales associated with sediment transport [6,19].

Management strategies must be based on scientific information. Monitoring data provide adequate tools to understand coastal processes and the analysis of what is happening during each period. Remote sensing techniques, which use satellite imagery to measure the morphology and scale of sandbars, have been introduced. Tracking techniques based on Landsat and Sentinel 2 images with the use of geographic information systems (GIS) and terrestrial surveys can be achieved to obtain indicators on coastal areas by using landmarks, global positioning system (GPS) light detection and ranging (LiDAR) and tri-dimensional (3D) scanners. However, the shoreline detection problem has still not been solved, since there are no algorithms that can be used without taking into account the objectives and characteristics of remote sensing images [36]. The resolution of the images has increased markedly. However, although the resolution has increased significantly, it is still not suitable for measuring small-scale dimensional changes. [31].

More recently, coastal zone surveys, including foreshore, backshore and dunes, have acquired and processed high-resolution data obtained by vertical take-off and landing (VTOL) drones, also known as UAV with cameras [37,38,39]. Research has been developed for monitoring with UAVs and it has allowed to obtain data and apply methodologies to integrate artificial vision algorithms for 3D and image processing.

The objectives and goals of the developed research for the region are to assess the state of coastal erosion by using integrated systems to provide a comprehensive and complete model for coastal erosion detection and monitoring and to promote a clear pathway that will serve for decision-making and coastal management.

## 2. Study Area

The Valencian oval is located on the Spanish Mediterranean coast (Figure 1) and registers significant coastal erosion. Due to coastal erosion, environmental problems and difficulties for the recreational and tourist uses of the coast have increased in recent decades. The Valencian coast has a total length of 470 km. Its central area is 256 km in length and extends from Ebro Delta to Cape San Antonio. The coastline is generally in the direction of S60° E.

The total population of the Valencian coast is 4.64 million inhabitants, and it has increased by almost 2 million in the last 50 years. The central region of Valencia extends to the north and south of the city and has a continuous conurbation of more than 2.9 million inhabitants. Currently, the central coastal area has a population density of 1238 inhabitants per square kilometer. It is an area with high tourist use and very important economic activity. Valencia’s port specializes in container traffic and has become a Mediterranean leader in this field, with the metropolitan area being a relevant logistic center. The effects of occupation and increased use, urbanization, degradation and destruction of coastal spaces, construction of port infrastructure and other infrastructure for land transport and hydraulic regulation have produced important sedimentary imbalances and an alteration of coastal dynamics with the consequent modification of coastal morphologies (e.g., beaches, deltas, dunes, mouths, etc.); these accumulations are the result of sedimentation and changes in profiles. Erosion in many coastal areas has increased due to the complex effects of climate change. Rising sea levels and changes in weather patterns have intensified the frequency of extreme events occurring on our planet [1,3,34,35,36].

An evaluation of the situation of the sedimentary balance, determining zones in accretion and others in erosion throughout the last 15 years, was obtained for the stretch of coast between the south port of Castellón and the port of Sagunto. Recently, to define interventions in areas with coastal erosion, topographical and bathymetric surveys have been developed in Almenara and the Sagunto beaches. This is a coastal area with extensive sandy beaches and dunes and is only interrupted by ports and small river mouths. Figure 2 shows the location of the coastal front where the monitoring tasks were developed. From north to south are: (a) Malvarrosa beach, between Queralt and Quartell, with a total length of 1200 m; (b) Corinto beach, south of Quartell, with a length of 1300 m; and (c) to the south, Almardà beach, with a coast length of 2030 m. The total length of the coast is 4530 m.

In September 2021, the stabilization works of the La Llosa and Almenara coastline began; these are based on a construction project that aims to stabilize the coastline in this area. The actions planned in this project focus on the beaches of La Llosa, Almenara and to the north of Sagunto, undertaking, among other things, the transfer of winnowed gravel from the Malvarrosa beach to the south of the Gola de Queralt, as well as the extension of jetties. From the point of view of fauna in the dune coastal area, it is worth mentioning the importance of the nesting *Charadrius alexandrinus*, a bird included in some catalogues of endangered fauna species in the vulnerable category. The coastal regeneration works have had to consider the nesting period in a very special way. For this reason, this project must stop during the spring period.

In order to assess negative effects on beaches, especially those located in the northern area of Almardà, Corinto and Malvarrosa beaches, the monitoring of coastal areas needs to consider whether or not there is coastal erosion as a result of these regeneration works on Almenara beach. The follow-up will focus on analyzing the evolution of the Sagunto beaches and coastal area. This monitoring was extended for three months, from November 2021 to January 2022.

## 3. Materials and Methods

This study aims to utilize a multidisciplinary approach by combining different analyses of maritime information, coastal processes and monitoring technologies to establish the causes of coastal erosion. An integrated analysis offers advantages in obtaining comparisons and simulations of dynamic situations in order to explore possible strategies to overcome erosion and, consequently, sustain public health and uses, economic growth, minimize population risk and maintain biodiversity in coastal ecosystems. The methodology in this study aims to achieve coastal erosion management and coastal erosion research, especially on vulnerable areas, which could be based on a risk assessment [14,34,37,38].

There is no periodic information on the geometry of the coast and its temporal evolution available. There are some assessments of advances and retreats of the coastline, but they are not systematic, nor do they take into account the position of the sea level and its effects; they are also not correlated with the maritime climate between each observation period. For this reason, it is necessary for the purposes of this study to correctly carry out these controls of the resulting geometry. To this end, determinations were made in the field through the use of surveys, which are described in detail later. The sources of information on coastal geometry and coastal sediments that were used have the methodological purpose of allowing comparisons to be made between different periods of time, so that the behavior and evolution of the beaches and the coastal area can be deduced, and sedimentary movements quantified.

To monitor the coastal–littoral area under investigation, it is necessary to develop a complete methodology [19,35] that allows for the analysis of, at least, the following information, so as to determine the following characteristics and their evolution over time:Sedimentary materials present on the beaches and contributions and movements made;Geometry of backshore, beach face and surf zone profile;Sea level in the period of observation and survey;Maritime climate data, especially having regard to wind and wave regimes in the monitoring period;Analysis of storms and their effects in the monitoring period;Anthropic actions taken, such as works, occupations, etc.

In addition, it was necessary to carry out work and analyze the granulometric characteristics of coastal materials, both native and those that may have been contributed. The collection of information in this regard must be as broad as possible, especially with regard to the implementation period. Surveys were performed over three months. For the purposes of monitoring and controlling the effects to be analyzed, continuous maritime information is being collected regarding the following:Sea level, through the analysis of tides provided by the Network of Tide Gauges (Red de Mareógrafos, REDMAR);Storm surges, based on prediction data, real-time records and an analysis of historical data from the State Ports Public Agency (Ente Público Puertos del Estado, EPPE) program through the link https://www.puertos.es/oceanografia/Paginas/portus.aspx (accessed on 2 December 2021).

### 3.1. Sediment Characteristics and Maritime Conditions

To obtain sediment characterization, which allows for the establishment of potential sediment sources to characterize coastal conditions in the study area, sediment samples were taken and analyzed in the laboratory. Sediment sampling was performed at a central point of each of the three beaches: Malvarrosa, Corinto and Almardá beaches (Figure 2). Granulometric tests of the sampled sediments were developed once a sample was washed and organic matter was removed using ASTM sieves. The carbonate contents of the samples were determined by removing it with acid. The mineralogical analysis of the samples was carried out by visual observation and counting of fractions of ASTM series by means of a microscope. The existence of blue quartz in the fractions was also obtained since this mineral is trace in that area of the origin of the sediments.

The sedimentary transport by waves and storm surges was estimated. The wave climate in the area was established [40,41,42,43,44] using different data sources. Tide data were obtained from Valencia tide gauge (REDMAR network) managed by Organismo Público Puertos del Estado (OPPE), which is an hourly data series provided by the Maritime Climate Program including data collected since 1992. To determine storm surge conditions, GOS 2.1 (Global Ocean Surges) reanalysis database [31] was used. The numerical model used in the GOS reanalysis is the ROMS (Regional Ocean Modeling System), a three-dimensional model developed by Rutgers’ Ocean Modeling Group (http://marine.rutgers.edu/po/index.php?model=roms, accessed on 9 November 2021). Relationships between the different sea reference levels were considered. Wind conditions (mean and extreme annual conditions) were determined, obtained through dynamic downscaling using the WRF-ARW 3.1.1 model (weather research and forecasting and advanced research dynamical solver) from the ERA-interim atmospheric reanalysis [40], developed by the European Center for Medium-Range Weather Forecasts (ECMWF). Wave data at deep water conditions were obtained, which are also from the GOW 2.1 (Global Ocean Waves 2.1 [44]). This reanalysis includes data collected since 1989 and has an hourly temporal resolution and a spatial resolution of 0.125° along the Mediterranean. The numerical model used for the generation of the reanalysis was the Wave Watch III model developed by the USA’s National Oceanic and Atmospheric Administration&National Centers for Environmental Prediction, NOAA/NCEP. The SWAN-OLUCA mixed numerical model for the wave propagation to the coastline was applied [44] to obtain the breaking currents.

Aerial and satellite views were also analyzed [43,45,46]. The longshore transport rate was estimated [47,48] using the following equation:Q (m^3^/year) = 2027 × 10^6^ × H_0_^5/2^ × sen(2α_0_) × cos(α_0_)^1/4^ × K_p_ × K_g_
where longshore transport rate, Q, is the amount of littoral drift to the right or to the left, past a point on the shoreline in an annual period. Wave conditions are computed with H_0_, significant wave height in deep water depths. Angle between wave crest and shoreline is α_0_. Coefficients, K_p_, K_g_, correct the probability of the wave height and direction, and the geometric configuration of the sector with the capacity to generate transport.

### 3.2. UAV System Data Collection and Zero Line

To monitor the geometry of the coast, the first task carried out was to establish control areas and points: stretches of coastline and profiles. Then, the downscaling of the information obtained from the Sentinel Hub satellite images, which is a cloud-based GIS platform for the distribution, management and analysis of satellite data, was addressed. Annual orthophoto images of the study area were obtained from the regional government of Valencia through the Institut Cartogràfic Valencià link, https://icv.gva.es/es/cartografia-tematica (accessed on 5 November 2021). All this allowed us to start the monitoring works.

For precision monitoring, field work was launched with the aim of carrying out monthly large-scale geometric surveys using drone systems. The processing of different images allows for remote sensing on shoreline detection and for the monitoring of coastal areas [36,37,39]. Applications of UAV technology can contribute to integrated coastal zone management (ICZM) and achieve the UN sustainable development goals (SDGs) [49]. The dynamic evolution of coastal areas in different areas has been researched by using UAVs [50,51,52,53]. In this study, a combination of different data is used to obtain an interdisciplinary approach.

For photogrammetric flights for geometric purposes, a global navigation satellite system (GNSS) base belonging to the air system was used to provide precise georeferencing in real time kinematic (RTK) to the images captured from the air. Flight planning was carried out in grid mode and adapted to the existing terrain in order to obtain homogeneous pixel size and precision.

The GNSS base of the aerial system is located in topographical bases. Precise XYZ coordinates were previously given to these bases, relying on the reference framework that defines the Spanish official reference system (which is the ETRS89 in UTM projection).

These coordinates were measured using a dual-frequency GNSS system with an accuracy of less than 1 cm.

In each flight, a series of support and control points (SACP) were also mediated with the dual-frequency GNSS system in order to provide precision guarantees. A precision report was delivered in each of the photogrammetric flights for the geometric analysis of the areas it has flown over. Flights were carried out with a professional drone (DJI Phantom 4 RTK and DRTK Base) and the GNSS system used to measure the SACP was a multi-constellation Spectra with a Trimble TDC600 controller (Figure 3). Once the areas to be flown over have been defined and located, flight planning was carried out for each area. This is performed using the drone’s own native software.

The planning parameters are to fly at a height relative to the ground of 36 m to obtain a pixel size or ground sample distance (GSD) of 1 cm/pixel, with the drone’s camera oriented completely from above so that the photo capture overlaps 80% longitudinally and 90% horizontally and transversely. Subsequently, the images captured by the drone are downloaded and processed using the drone’s native software to enable precise georeferencing with respect to the data measured by the drone’s GNSS system. This was performed post-process, whereby the position given by the GNSS base in the fraction of a second that the camera captured each of the aerial images was synchronized.

Next, Pix4D software is used to process all the georeferenced images in order to obtain the necessary metric products. These are the 3D Point Cloud, orthophotography and digital terrain model (DTM). From these products, other cartographic products can be obtained ex post in any format, including both raster and vector.

Finally, the zero line (ZL) is defined along the entire coastline using GPS methodology. The study area does not have a significant tide. Small time scale sea level changes are mainly due to wind set-up, storm surges and wave conditions. The swash zone forms the land–ocean boundary at the landward edge of the surf zone, where waves runup and rundown the beach face. The ZL is defined by runup and rundown interpolation in the swash zone. This method allows us to precisely define the edge of the beach. It can be complemented with the correction of the sea level by astronomical or meteorological tides and wave oscillation. The data collection is extended up to the −1.00 level in profiles taken between 25 m. Consequently, detailed information can be obtained from the most exposed and dynamic sections of the submerged beach profile. The total length of the stretch of coast to be monitored with the ZL is 4530 m long. All the obtained information must be analyzed throughout the monitoring time with information on the maritime climate.

## 4. Results

The northern part of the Spanish Mediterranean coastline is generally in the direction of S60°E; it is fairly straight and low-lying without natural indentations. There are a large number of port infrastructure and coastal protection systems, including jetties, dykes and breakwaters. It suffers considerable erosion. The coast, especially in the Valencian oval, registers significant erosion due to the north–south movement of materials and the absence of sediments from the north, which is mainly due to the presence of docking installations and other construction. Beaches are mostly composed of D_50_ = 0.25 mm sand and occasionally gravel close to river mouths. Many bathymetric studies have detected submarine materials, mainly composed of fine sand of D_50_ = 0.11–0.15 mm [49].

Problems of erosion have been occurring in recent decades. The effects of occupation and increased use, urbanization, degradation and destruction of coastal spaces, construction of port infrastructure and others for land transport and hydraulic regulation, etc., have produced significant sedimentary imbalances. The Valencian oval coast, particularly between the port of Castellón and the port of Valencia, has intensively recorded these effects. Along this coast, longitudinal protection, groins, detached breakwaters and many artificial beach nourishments have been built. All these structures, especially the ports and groins, have produced significant effects on littoral transport rates.

On the northern beaches, especially in Almenara and the Malvarrosa beaches, the fine sands have disappeared due to the action of littoral transport, which makes bathing on this beach difficult during the tourist period, especially for elderly people. Sediments in the surf zone comprise 51.3% gravel. On the beaches of Corinto and Almardá, the gravel percentages vary between 5.3 and 22.1%. The D_50_ of sediments on these beaches varies between 0.19 and 0.29 mm (Figure 4). For comparison, results of the mean sediment sizes on these beaches in 2007 and 2008 can be seen in Table 1.

In September 2021, sediment samples were taken in two sections of Malvarrosa beach, 1 and 2, (Figure 4). In each section, samples were obtained on the coastline (1a and 2b), at 1 m above sea level on the beach (1c and 2d), and at 2 m above sea level on the beach (1e and 2f). Results of the granulometric analysis of these samples can be seen in Table 2, where D_50_ size, percentages and types of different sediments can be seen.

Research has been carried out based on aerial photos, which determined the volumes of additions and losses of sediments on the coastline. Figure 5 shows statistics maritime conditions considered to estimate longshore transport rates. Black line in Figure 5 represents registered conditions. Figure 6 shows the evolution of the coastline, calculated by photogrammetry, between Almenara and Sagunto. Evolution of the coastline is represented in different periods with reference to the original line, the coastline, in 1947. The Malvarrosa beach shows greater erosion, which decreases towards the south in the Corinto and Almardá beaches. The Almardà beach registers accumulations. The coastline of the Almardá beach was extended between 1994 and 2000.

At Corinto beach, in one 1.4 km stretch a volume of 820,152 m^3^ accumulated between 1957 and 1965, and there was an erosion of 1340,584 m^3^; therefore, it can be concluded that the transport capacity was approximately 65,000 m^3^/year. During the analyzed period, the detected trend is the recession in the whole section. There are more pronounced variations at the Malvarrosa and Corinto beaches, while at Almardà, the variation is very uniform with a tendency for accretion or apparent stability.

Figure 7 represents the rate of evolution. This rate has an average annual value of 33 cm (−0.33 m) for the entire coastal area. The beaches of Malvarrosa and Corinto La Costa suffer from erosion, while Almardà shows a slight tendency towards accumulation. All this is clearly due to the north–south direction of net littoral solid transport.

To monitor the geometric changes in the coast, three campaigns with UAV were carried out, one per month, between November and December 2021 and January 2022. Figure 8 shows the areas where campaigns were developed. Figure 9 shows some UAV image results in these areas and campaigns of the total monitored stretch of coast. In these areas there are no significant changes in sea level due to tides. The changes recorded in the campaign period involved variations in sea level with a maximum amplitude of 12 cm due to weather conditions. Consequently, the influence of this amplitude of sea level changes in the width of the beaches is not relevant. The slope of the swash zone has an average slope of 34%. Consequently, variations in sea level of 12 cm imply maximum width changes of 35 cm.

In the study area, it was possible to detect through monitoring an annual erosion of 12 m on Malvarrosa (Figure 10) beach and 6 m on Corinto beach. This evolution of the beaches can be defined as Class 6-High Erosion [7]. This definition of the beach evolution classes establishes seven classes, from Class 1-Very High Accretion to Class 7-Very High Erosion. The monitoring program has evidenced the shoreline trend as a result of the impact of rigid structures, mainly ports and groins, in promoting down-drift erosion processes in the area.

## 5. Discussion

Cities and urban areas expand their spaces, and the services they manage converge with the management of coastal spaces and necessary preservation. This has produced the alteration of coastal dynamics with the consequent modification of coastal morphologies (e.g., beaches, deltas, dunes, mouths, etc.), erosion, accumulations due to sedimentation, changes in profiles, etc. The vulnerability of the coast is rapidly increasing due to climate change. Vulnerability is manifested via increases in flood levels, beach profile changes, erosion, increases and changes in coastal morphology, variations in littoral transport rates and coastal erosion. Ports can create coastal erosion by altering wave patterns. The research that was carried out attempts to clarify environmental effects in managing port-induced coastal erosion occurring at beaches, which are extensively used by the population.

Changes in weather patterns due to climate change produce an intensification of the frequency of extreme sea events. Beaches in this area are showing coastal sedimentary movements that lead to the appearance of gravel and its propagation towards the south in very significant magnitudes. This has been very noticeable after recent extraordinary storms in September 2019 and the Gloria storm in January 2020.

On the Spanish Mediterranean coast, more than 90% of the sand and gravel comes from rivers, torrents and streams, while the rest comes from erosion materials from cliffs, in addition to vegetable matter. Granulometric results show that sediment beaches along the Sagunto and Almenara coast are a mix of sand and gravel. Until a few years ago, Almenara and Sagunto beaches maintained a sand cover, giving the impression of sandy beaches. Recently, erosive processes in this coastal area have displaced the sand and exposed the gravel (Figure 11).

The obtained shoreline change rates show that problems in the coast between the ports of Castellón and Sagunto are the result of general degradation of natural conditions, resulting in a fragmented coast with sedimentary imbalances generated by ports and other infrastructure, human use and invasive urbanization of coastal spaces. River regulation and sediment reduction supplies to the coast and the effects of climate change have increased erosion. At the same time, to correct the situation, there have been no investments made according to environmental, natural and socioeconomic values. ICZM under the principle of sustainability is needed. The current result is a degraded and fragmented coast, which requires management and intervention through coordinated action and sustainability.

Over the years, the imbalance in sediment and transport, which is strongly influenced by port constructions and groins, has been sought to be mitigated through the building of various defense works. The entire area under investigation here has been altered from its natural initial dynamic. The areas with the greatest problems are those in which strong transport and urban areas coincide. The current situation is attributable to successive transformations that have taken place over time in an initially continuous coast, with an area of lagoons and incipient dunes. The construction of the ports of Sagunto in 1902, Castellón in 1915 and Burriana in 1932 reduced littoral transport rates, and a significant volume of sediments were retained upstream of these ports. The urbanism developed over the years, which was more intense from the 1960s/1970s and affected natural conditions, especially in dune areas. Solutions given to these problems were local, with coastal defenses mostly developed in front of urban centers, which gave rise to even greater sedimentary imbalances, forming important concavities downstream (Figure 12).

The diagnosis of sections between the port of Burriana and the port of Sagunto recommend a reanalysis study on all the currently existing defense works, taking into account climate change and extending to critical points with serious current erosion. With the start-up of the stabilization works on the coast of Almenara and Sagunto, it was possible to develop a monitoring system that provides data on coastal evolution on an appropriate time and space scale. Dunes and beach changes were evaluated. Coastal management requires us to define its characteristics and evolution. The task of continuously collecting information is a basic necessity for proper management and, in addition, today there are increasingly powerful and accessible technological means for it.

ICZM is a key element for the sustainable development of the coastal areas. One of the consequences of climate change is the increase in global coastal erosion and the loss of valuable coastal areas. A large number of development activities, such as construction of ports, urbanization, massive tourism, overexploitation of aquifers and the presence of dams, are increasing the vulnerability of coastal areas. Due to extraordinary storms on the coast, there is a great urgency for ICZM. Action must be addressed with a robust system based on coastal monitoring and the evaluation of results. Recent trends in coastal erosion mitigation are addressing towards soft and innovative methods, since hard methods have important impacts such as high cost and down-drift erosion. The debate on coastal sustainability must consider the effectiveness and adequacy of necessary investments, consolidating them into conservation plans and ICZM.

## 6. Conclusions

To act against coastal erosion and correct the current situation affecting dunes and beaches, it is necessary to promote soft and innovative solutions such as sand bypassing, dune rehabilitation and dune vegetation. These solutions will also work to prevent coastal erosion due to stronger storms caused by climate change. During the stabilization works of a coastal area on the Spanish Mediterranean coast, a monitoring investigation was developed based on the use of UAVs and maritime and sediment information. Coastal monitoring provided appropriate information because of its spatial and temporal dimensions and data quality to strengthen scientific knowledge and sustainability.

This monitoring program made it possible to manage very precise information, and high-resolution images can be processed to detect changes, both from the works in progress and from natural changes due to the action of winds, waves and tides. This monitoring program is a good tool for making decisions that must be based on scientific information and takes advantage of technological development with very low economic costs.

## Figures and Tables

**Figure 1 ijerph-19-05457-f001:**
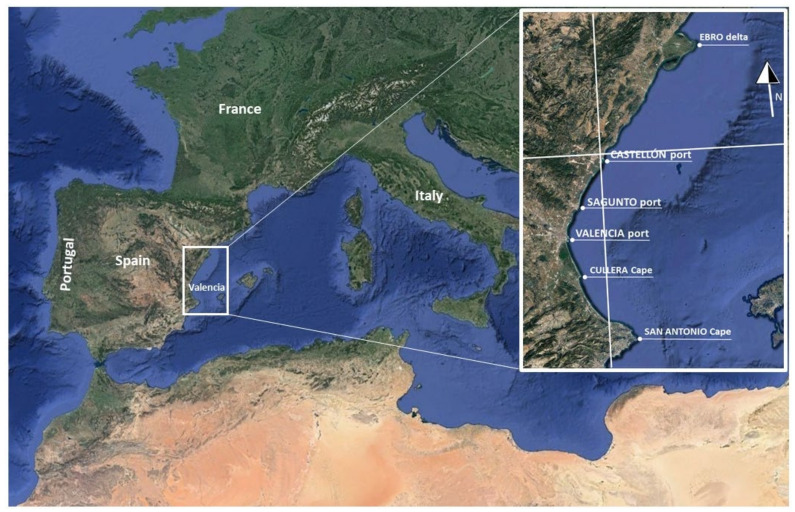
Valencian oval location (prepared by the authors based on Google Earth).

**Figure 2 ijerph-19-05457-f002:**
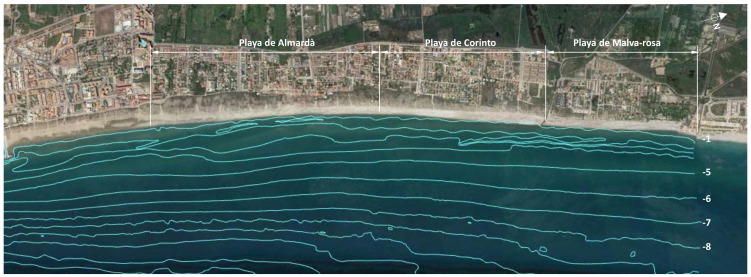
Bathymetry in the Almenara–Sagunto coastal area (prepared by the authors based on Google Earth).

**Figure 3 ijerph-19-05457-f003:**
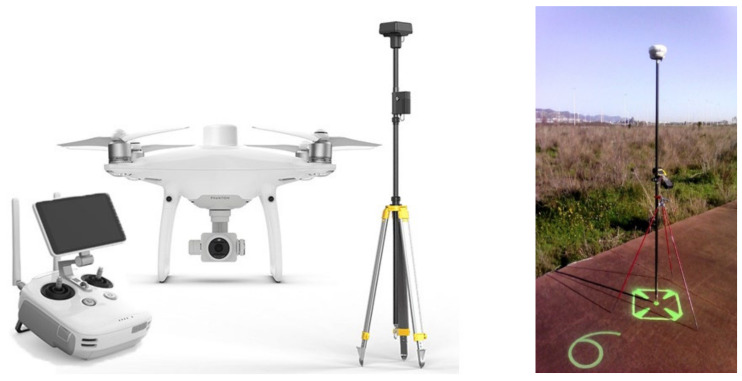
The drone and GNSS base used for photogrammetric flights (provided by General Drones, SL).

**Figure 4 ijerph-19-05457-f004:**
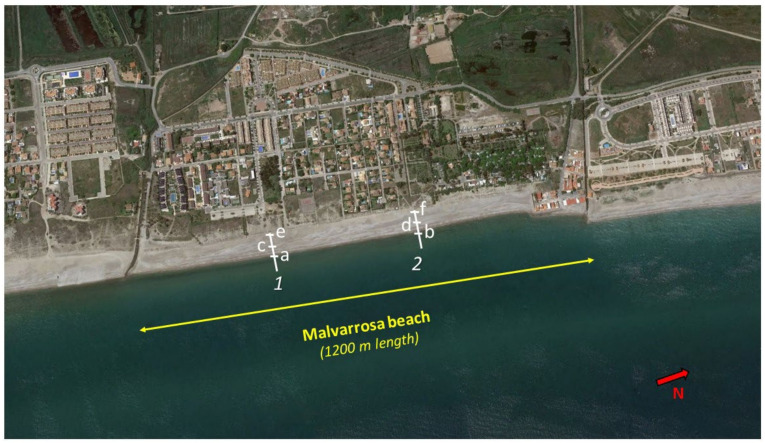
Section and point sediments sampling in Malvarrosa beach (prepared by the authors based on Google Earth).

**Figure 5 ijerph-19-05457-f005:**
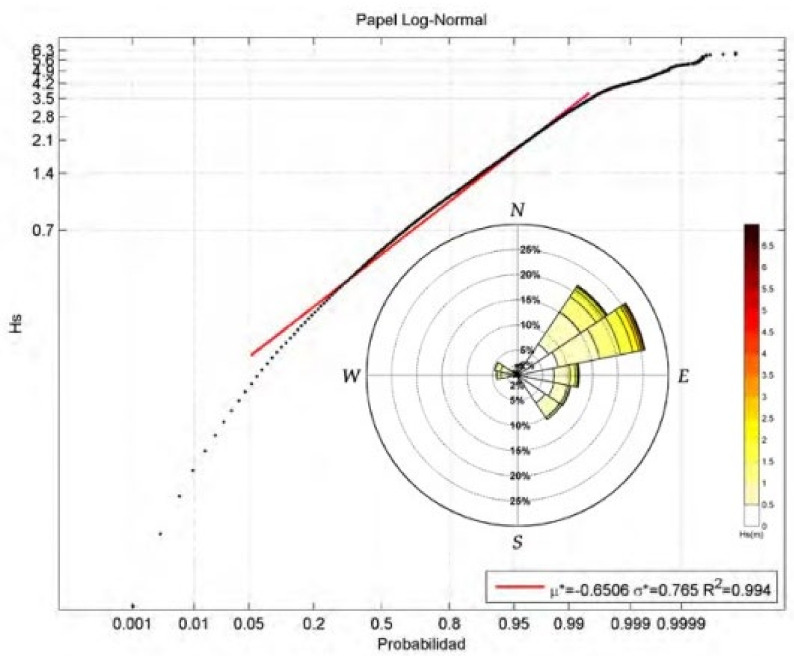
Mean scalar regime and significant waves in deep water coastal area of Almenara–Sagunto, showing relation between significant wave height in deep water depths, Hs, and probability of non-exceedance in *x*-axis (Instituto de Hidráulica Ambiental de Cantabria).

**Figure 6 ijerph-19-05457-f006:**
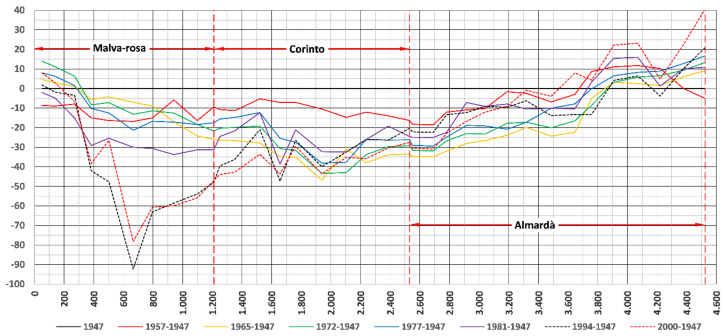
Shoreline changes in meters (*y*-axis) and coastal evolution in Malvarrosa, Corinto and Almardá beaches (*x*-axis, distance from the north end of the section studied to the south) (prepared by the authors).

**Figure 7 ijerph-19-05457-f007:**
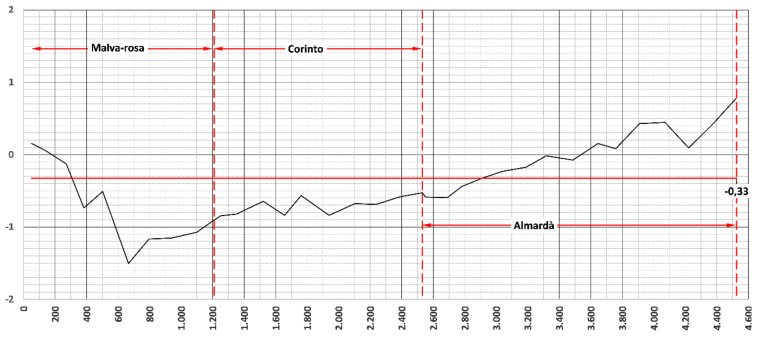
Coastal evolution rates at Malvarrosa, Corinto and Almardá beaches referring to 1947 situation (*x*-axis, distance from the north end of the section studied to the south) (prepared by the authors).

**Figure 8 ijerph-19-05457-f008:**
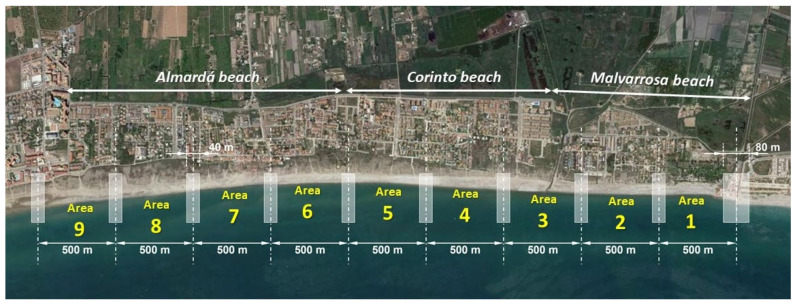
Different coastal areas considered for monitoring works at Malvarrosa, Corinto and Almardá beaches (prepared by the authors).

**Figure 9 ijerph-19-05457-f009:**
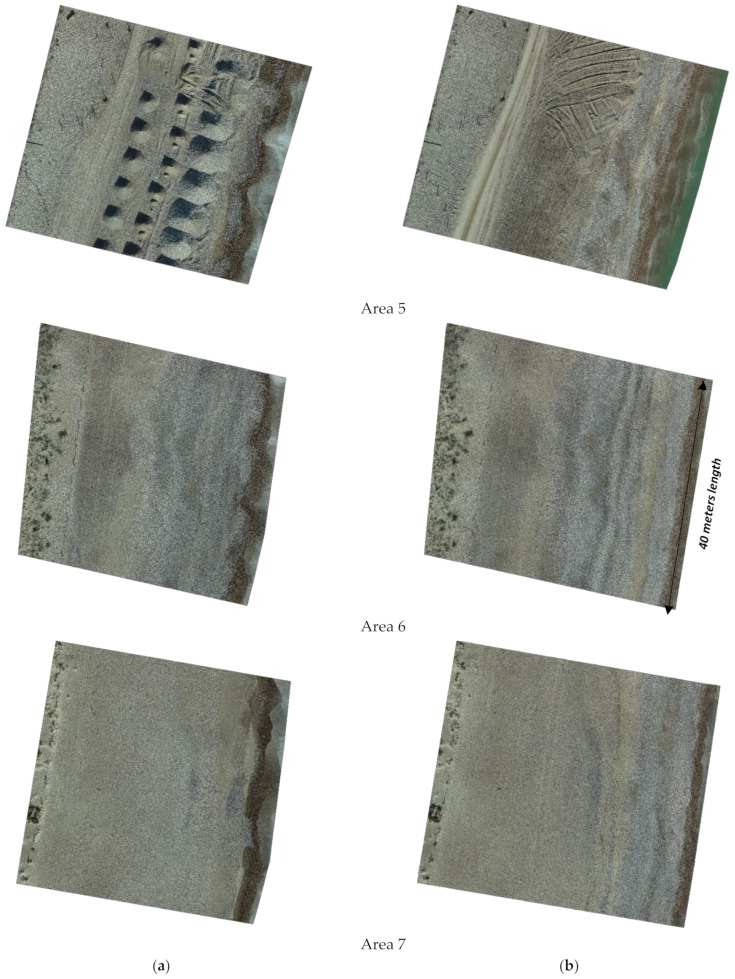
Different coastal area images in different campaigns showing changes and short-term coastal evolution at Malvarrosa, Corinto and Almardá beaches: (**a**) November 2021; (**b**) January 2022. Images are rotated to show the north facing upwards (prepared by the authors).

**Figure 10 ijerph-19-05457-f010:**
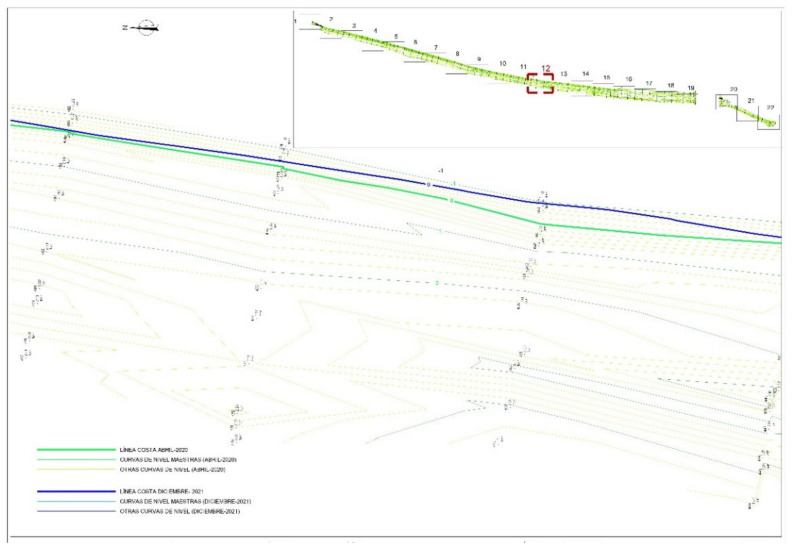
Evolution of coastline in Malvarrosa beach since April 2021 (green line), to December 2022 (blue line) (prepared by the authors).

**Figure 11 ijerph-19-05457-f011:**
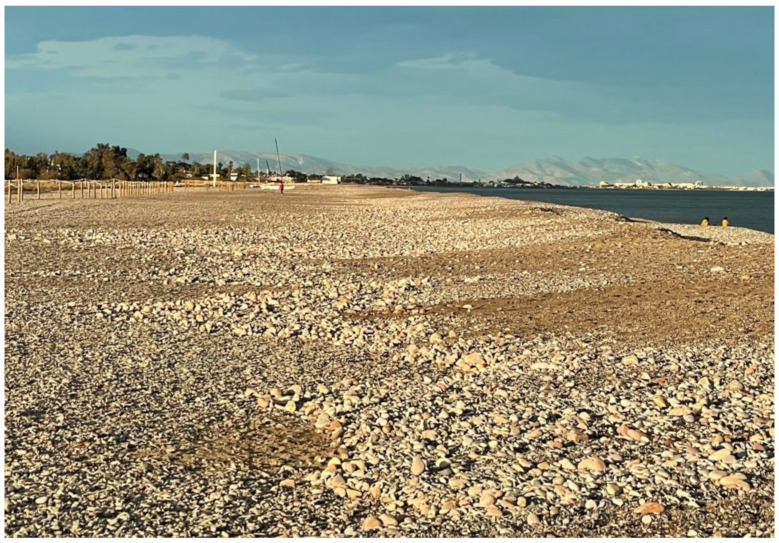
December 2021, image of Malvarrosa beach facing northward. Coastal erosion has caused the disappearance of the sand cover. Coarse sediments, such as gravel, remain on the beach.

**Figure 12 ijerph-19-05457-f012:**
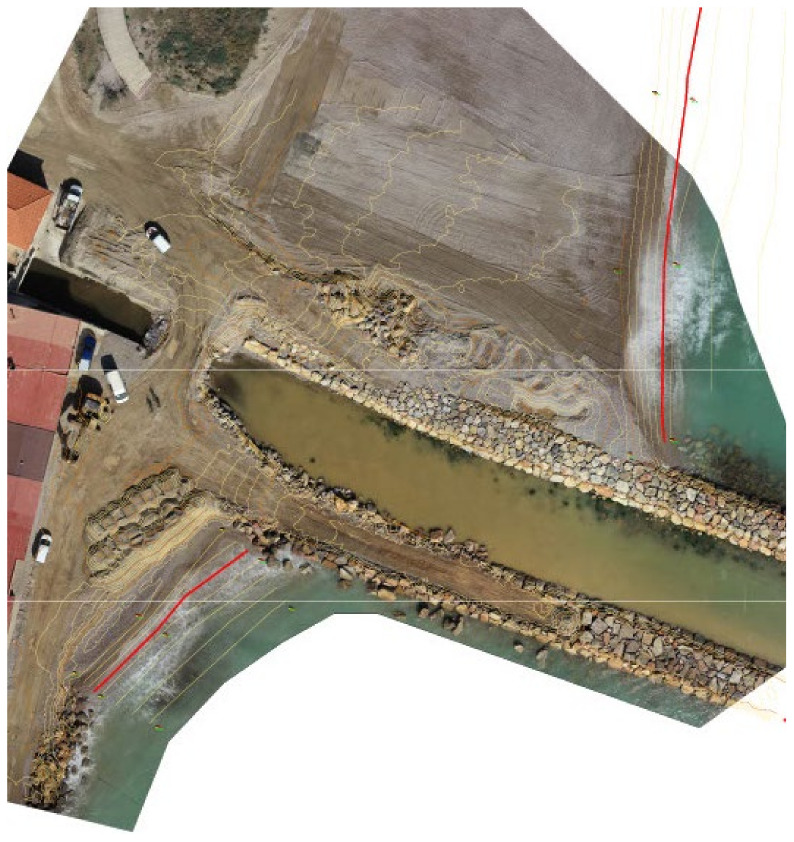
Orthophotography of defense coastal erosion works at Almenara and Sagunto beaches; ZL (red lines) and DTM (yellow lines) from UAV.

**Table 1 ijerph-19-05457-t001:** Granulometric D_50_ size results in November 2007 (A) and September 2008 (B) [41].

Almardá Beach			Corinto Beach		
Height/Depth	A D_50_ (mm)	B D_50_ (mm)	Height/Depth	A D_50_ (mm)	B D_50_ (mm)
2	0.21	0.22	2	0.30	0.24
1	0.50	0.25	1	>2	>2
0	1.80	0.30	0	>2	>2
−1	0.14	0.18	−1	0.19	0.20
−2	0.19	0.16	−2	0.13	0.17
−3	0.12	0.14	−3	0.11	0.14
−4	0.12	0.13	−4	0.17	0.13
−5	0.18	0.12	−5	0.13	0.15
−6	0.16	0.12	−6	0.11	0.16
−7	0.19	0.11	−7	0.11	0.15
−8	0.11	0.11	−8	0.18	0.15

**Table 2 ijerph-19-05457-t002:** Granulometric analysis, D50 size, % and type of sediments in Malavarrosa beach September 2021Malvarrosa beach.

ASTM nr	Sieving Size (mm)	Height (in Meters) of the Beach Sample Collection					
		Shoreline		+1		+2	
		1a	2b	1c	2d	1e	2f
		% Retained		% Retained		% Retained	
2 ^1/2^	63	0	0	0	0	0	0
1	10	0	0	0	0	0	0
3	6.73	0	0	0	0	0	0
5	4	0.3	4.6	3.8	7.3	0.3	0.2
10	2	6.4	17.2	3.5	8.5	0.1	1.0
18	1	12.1	20.4	4.3	12.3	0.2	5.8
25	0.717	9.7	9.8	6.3	6.7	2.8	11.0
40	0.425	20.2	19.3	25.9	16.3	16.4	15.0
80	0.180	25.5	16.7	31.7	27.7	41.2	27.1
120	0.125	20.9	11.1	22.2	20.4	34.6	13.0
200	0.075	4.9	0.9	2.1	0.7	4.0	2.5
>200	<0.075	0	0	0.2	0.1	0.4	0.1
D_50_ (in mm)		0.38	1.04	0.25	0.34	0.27	0.32
% gravel		6.7	21.8	7.3	15.8	0.4	1.2
% sand		93.3	78.2	92.5	84.1	99.2	98.6
% silt and clays		0	0	0.2	0.1	0.4	0.2
Beach material classification		sand	sand and gravel	sand	sand	sand	sand

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
