# Peer review of "Coastal Monitoring Using Unmanned Aerial Vehicles (UAVs) for the Management of the Spanish Mediterranean Coast: The Case of Almenara-Sagunto"

_ijerph, 2022, doi:10.3390/ijerph19095457_

Round 1
Reviewer 1 Report
This study aims to evaluate erosion along the Valencian Oval using various techniques for monitoring and measuring of sedimentary characteristics, environmental conditions, and maritime information. The authors develop a strong approach for monitoring in this region and use a series of effective monitoring tools and techniques including sediment sampling, UAVs and drone imagery, maritime data, and site evaluations. Together, these data offer important lessons on erosion and sedimentary transport in the region. There is certainly merit to their approach and analysis, however, there are several areas in the paper that do not adequately address or describe details of the study, its methods of data collection, or necessary discussion of results. I have also suggested some reorganization of the study objectives to focus on the primary aims of this research in order to add clarity to the paper. Below, I have provided comments and suggestions for the authors to review. Many of these revisions I consider only minor, however, some are critical to the paper and should be reviewed carefully.
Abstract
Include some details of your findings and conclusion.
Introduction
40: capitalize I in “in” at start of new sentence.
46: “city and coastal agglomeration (CAC)” Does CAC = “city and coastal”? The acronym doesn’t seem to fit here. Consider revising.
48-50: This sentence is repeated from 40-42.
57: IZCM is not defined.
68-69: The authors indicate that “The fundamentals of sedimentary transport have been studied extensively”. But they only include a single reference. It is relevant, but perhaps you can indicate the “extensive” research that has been completed with a selection of studies, i.e. a few major ones.
81-86. Long, run-on sentence. Consider revising for clarity.
95-103: Objectives and goals.
I understand the aim here is to establish a method for monitoring coastal areas. But I feel like the first two objectives are not necessary to include. In my view, it is implied that you have developed a method and are using it to collect data, analyze, and monitor. This is indicated in your last objective “to promote a clear pathway that will serve for decision-making and coastal management”. I think it would help to focus on the second two objectives which are your main focus.
*developing a methodology is not the objective of this study. If it is, then the authors need a clear discussion of this in the results and discussion sections. But since the tools the authors are using have already been developed, there is no need to create a study objective to test the methods. You can use the established method to study ICZM, etc.
Also, the formatting is awkward with the dash marks. Consider writing out in sentence form.
Study Area
106: “Problems have increased in recent decades” does not need to be included in the methods description. And if it remains, explain what problems you are referring to. It’s too vague.
Figure 1: My only suggestion is that you place a border around your expanded section of the map. It seems to blend in with the rest of the map so this will help the reader more quickly focus on the expanded area.
125: “A general evaluation […]”
126: “Intensive surveys have been developed […]”
Can the authors be more specific? These terms are too vague and more information is warranted.
134-145: The section about nesting birds does not seem so important to the main objectives of your work. Did this really inhibit your study? My suggestion is to shorten this to 1-2 sentences if it has not affected your study. (related to this - see my discussion comment)
151: Can you provide a clear definition of the “zero line”. Or, I suggest, avoid using this term until you introduce it in your methods section.
Methods and Materials
I suggest shortening the open section (lines 156-198).
For instance, the first paragraph opens with “This study aims […]”, but this is not the place to discuss aims or objectives or rationale for this study. I think it would be helpful to get right to the methods/approach and this will allow the authors be more concise. Consider this a suggestion.
204-209: The authors are listing the methods used to characterize sediments. But it lacks important detail. Can you explain, in more detail, where you collected samples, how you determined carbonates and minerology. In my opinion, point form is not an effective way to describe the methods here.
Maybe this data best belongs in a table outlining the parameters you are measuring and the various methods of data collection.
Same issue with the transport rate calculation. It’s just not clear from the point form. Discuss in detail the data used and the calculation performed.
234: Longshore transport rate equation. Define the variables.
Same issues with point form in UAV system data collection and zero line section
Can you provide a clear definition of “zero line”. 287-289 only provides information on how it was collected.
244: “Analysis of the information data from monitoring the works.” I am not sure what this refers to. Explain further.
Figure 3: Are there any copyright concerns with using the stock photo on the left (if it is a stock photo)?
Results and discussion:
The first section in results (like the M&M) doesn’t seem to be relevant to this section. I would pass over the initial overview/background stuff and get right to the results. The rationale and background were already provided.
302: You state a population of 5 million. But on line 109 you indicate a population of 4.64 million. I suggest be consistent.
306-307: What does “a relevant tourist development” mean? Is this referring to the tourism industry? Perhaps be more specific.
Figure 4: This is very busy and the text is too small and blurry. I can’t interpret nearly any of the data here. Is this data repeated in Table 1?
I suggest you make a single figure for the data and include the remaining tables/information in the SI.
Figure 5: Define Hs. Define x-axis.
Figure 6: Define axes, units. Perhaps in the figure caption explain that these are relative values, i.e. difference from 1947.
Figure 7 title suggests this is the same data as in Fig 6. But, of course, the data is different. Perhaps change the title or somehow indicate the relevance of fig 7. Add axes titles and units.
What year was this measured? I.e. what does the green line refer to?
Figure 8: Can you define area 5, 6, 7? Perhaps indicate that each column is a different site, and each row is a different campaign. Is there any rationale to why these images are rotated? Perhaps to fit directionally north. It’s difficult to interpret these for the reader with an untrained eye. Consider making these figures smaller, closer together.
What is the scale of these photos?
Figure 9: This is not a very effective caption for this photo. This is a discussion point. Consider revising.
379-380: “The significance of this variation on beach dimension is minor”
How did you determine the degree of this variation? What calculations or statistical analyses were performed?
The discussion section lacks context on the importance of ICZM and minimizing erosion. Can you provide more discussion relating to the significance of ICZM under the principle of sustainability? Consider the social, economic, ecological effects if this continues. Perhaps you can frame this using the example of nesting birds. Will they have areas to nest if erosion continues to transport sand.
Author Response
Reply to Reviewer 1
The manuscript has been revised following your comments and indications. We appreciate the very detailed comments and suggestions that you have been noted. We believe that we have modified the original manuscript in a very important way, describing details of the study, reorganizing some sections and correcting deficiencies that you had been pointed out. In particular:
“This study aims to evaluate erosion along the Valencian Oval using various techniques for monitoring and measuring of sedimentary characteristics, environmental conditions, and maritime information. The authors develop a strong approach for monitoring in this region and use a series of effective monitoring tools and techniques including sediment sampling, UAVs and drone imagery, maritime data, and site evaluations. Together, these data offer important lessons on erosion and sedimentary transport in the region. There is certainly merit to their approach and analysis, however, there are several areas in the paper that do not adequately address or describe details of the study, its methods of data collection, or necessary discussion of results. I have also suggested some reorganization of the study objectives to focus on the primary aims of this research in order to add clarity to the paper. Below, I have provided comments and suggestions for the authors to review. Many of these revisions I consider only minor, however, some are critical to the paper and should be reviewed carefully.”
We thank you for the general comment. According to your suggestion we have reorganized some parts. We understand and hope that we have added clarity to the manuscript.
Abstract
Include some details of your findings and conclusion.
Done
Introduction
40: capitalize I in “in” at start of new sentence.
Done
46: “city and coastal agglomeration (CAC)” Does CAC = “city and coastal”? The acronym doesn’t seem to fit here. Consider revising.
We have changed the acronym. We have used CCA
48-50: This sentence is repeated from 40-42.
We have removed the repetition and rewritten
57: IZCM is not defined.
Acronym has been explained before use and we have changed it. The correct acronym to refer to the integrated coastal zone management is ICZM
68-69: The authors indicate that “The fundamentals of sedimentary transport have been studied extensively”. But they only include a single reference. It is relevant, but perhaps you can indicate the “extensive” research that has been completed with a selection of studies, i.e. a few major ones.
We have rewritten this section. Additional information has been added.
81-86. Long, run-on sentence. Consider revising for clarity.
Sentence has been rewritten. Thank you so much for your comment, you were absolutely right
95-103: Objectives and goals.
I understand the aim here is to establish a method for monitoring coastal areas. But I feel like the first two objectives are not necessary to include. In my view, it is implied that you have developed a method and are using it to collect data, analyze, and monitor. This is indicated in your last objective “to promote a clear pathway that will serve for decision-making and coastal management”. I think it would help to focus on the second two objectives which are your main focus.
*developing a methodology is not the objective of this study. If it is, then the authors need a clear discussion of this in the results and discussion sections. But since the tools the authors are using have already been developed, there is no need to create a study objective to test the methods. You can use the established method to study ICZM, etc.
Also, the formatting is awkward with the dash marks. Consider writing out in sentence form.
Your indications have been attended. These paragraphs have been reorganized. Some of them have been removed. Some additional explanations have been added
Study Area
106: “Problems have increased in recent decades” does not need to be included in the methods description. And if it remains, explain what problems you are referring to. It’s too vague.
Paragraph have been reorganized. Some additional explanations have been added
Figure 1: My only suggestion is that you place a border around your expanded section of the map. It seems to blend in with the rest of the map so this will help the reader more quickly focus on the expanded area.
Done
125: “A general evaluation […]”
126: “Intensive surveys have been developed […]”
Can the authors be more specific? These terms are too vague and more information is warranted.
Additional explanations have been added
134-145: The section about nesting birds does not seem so important to the main objectives of your work. Did this really inhibit your study? My suggestion is to shorten this to 1-2 sentences if it has not affected your study. (related to this - see my discussion comment)
Following your suggestion the sentences have been shortened
151: Can you provide a clear definition of the “zero line”. Or, I suggest, avoid using this term until you introduce it in your methods section.
Following your suggestion the sentences concerning “Zero line” has been eliminated. Zero line definition is established in section 3.2: “The study area does not have a significant tide. Small time scales sea level changes are mainly due to wind set-up, storm surges and wave conditions. The swash zone forms the land-ocean boundary at the landward edge of the surf zone, where waves runup and rundown the beach face. The ZL is defined by runup and rundown interpolation in the swash zone.”
Methods and Materials
I suggest shortening the open section (lines 156-198).
For instance, the first paragraph opens with “This study aims […]”, but this is not the place to discuss aims or objectives or rationale for this study. I think it would be helpful to get right to the methods/approach and this will allow the authors be more concise. Consider this a suggestion.
We believe that the introduction of this section supports the methodology used. In any case, we appreciate the suggestion.
204-209: The authors are listing the methods used to characterize sediments. But it lacks important detail. Can you explain, in more detail, where you collected samples, how you determined carbonates and minerology. In my opinion, point form is not an effective way to describe the methods here.
Maybe this data best belongs in a table outlining the parameters you are measuring and the various methods of data collection.
Same issue with the transport rate calculation. It’s just not clear from the point form. Discuss in detail the data used and the calculation performed.
Explanations detailing what was requested have been incorporated. Figures have been added. This section has been extensively reorganized following your comments and suggestions.
234: Longshore transport rate equation. Define the variables.
We have defined them. Done
Same issues with point form in UAV system data collection and zero line section
Your indications have been attended. These paragraphs have been deeply reorganized. Some of them have been removed. Some additional explanations have been added
Can you provide a clear definition of “zero line”. 287-289 only provides information on how it was collected.
We have defined it. Done
244: “Analysis of the information data from monitoring the works.” I am not sure what this refers to. Explain further.
This section has been reorganized and rewriten.
Figure 3: Are there any copyright concerns with using the stock photo on the left (if it is a stock photo)?
No copyright applies
Results and discussion:
The first section in results (like the M&M) doesn’t seem to be relevant to this section. I would pass over the initial overview/background stuff and get right to the results. The rationale and background were already provided.
First section has been removed. You are right
302: You state a population of 5 million. But on line 109 you indicate a population of 4.64 million. I suggest be consistent.
This part has been moved to another section and the data has been corrected. Indeed, there was inconsistency in the previous manuscript.
306-307: What does “a relevant tourist development” mean? Is this referring to the tourism industry? Perhaps be more specific.
This sentence has been rewriten. Additional explanations have been added
Figure 4: This is very busy and the text is too small and blurry. I can’t interpret nearly any of the data here. Is this data repeated in Table 1?
I suggest you make a single figure for the data and include the remaining tables/information in the SI.
You were absolutely right. That figure has been removed. The data that figure showed have been incorporated into a table that allows you to see all the necessary details. See the new Table 2. Granulometric analysis, D50 size, % and type of sediments in Malavarrosa beach September 2021
Figure 5: Define Hs. Define x-axis.
Done
Figure 6: Define axes, units. Perhaps in the figure caption explain that these are relative values, i.e. difference from 1947.
Done. New explanation has been added in figure caption. Te proposed caption is: Shoreline changes in meters (y-axis)and coastal evolution in Malvarrosa, Corinto and Almardá beaches (x-axis, from the north end of the section studied to the south)
Figure 7 title suggests this is the same data as in Fig 6. But, of course, the data is different. Perhaps change the title or somehow indicate the relevance of fig 7. Add axes titles and units.
What year was this measured? I.e. what does the green line refer to?
Coastal evolution rates at Malvarrosa, Corinto and Almardá beaches referred to 1947 situation. This is the new text in figure caption
Figure 8: Can you define area 5, 6, 7? Perhaps indicate that each column is a different site, and each row is a different campaign. Is there any rationale to why these images are rotated? Perhaps to fit directionally north. It’s difficult to interpret these for the reader with an untrained eye. Consider making these figures smaller, closer together.
What is the scale of these photos?
We have added a new scheme in the figure showing the location of these áreas. Scale has been added. And, following your suggestion we have made figures smaller. You were right and we believe that the final result is better. We have also explained that images are rotated to show the nord facing upwards.
Figure 9: This is not a very effective caption for this photo. This is a discussion point. Consider revising.
Figure caption has been rewritten
379-380: “The significance of this variation on beach dimension is minor”
How did you determine the degree of this variation? What calculations or statistical analyses were performed?
We have rewritten this sentence and explained result. the influence of this amplitude of sea level changes in the width of the beaches is not relevant. The slope of the swash zone has an average slope of 34%. Consequently, variations in sea level of 12 centimeters imply maximum width changes of 35 centimeters.
The discussion section lacks context on the importance of ICZM and minimizing erosion. Can you provide more discussion relating to the significance of ICZM under the principle of sustainability? Consider the social, economic, ecological effects if this continues. Perhaps you can frame this using the example of nesting birds. Will they have areas to nest if erosion continues to transport sand.
Discussion section has been also rewritten: ICZM is a key element for the sustainable development of the coastal areas. One of the consequences of climate change is the increase in global coastal erosion and the loss of valuable coastal areas. A large number of development activities like construction of ports, urbanization, massive tourism, overexploitation of aquifers and the presence of dams, is increasing the vulnerability of coastal areas. Due to extraordinary storms on the coast, there is a great urgency for ICZM. Action must be addressed with a robust system based on coastal monitoring and the evaluation of results. Recent trends in coastal erosion mitigation is addressing towards soft and innovative methods, since hard methods have important impacts as high cost and down-drift erosion.
Thank you again for your comments and suggestions.
Reviewer 2 Report
Introduction: I recommend rewriting the whole introduction considering that you should explain:
- the reasons behind this study in view of the literature gaps;
- if this study is an integrated approach of both physical, administrative, biological parameters, you should compare it with the most faculty studies on this field (there are dozens of studies about coastal management indicators and integrated approaches; see for example:
Cabezas-Rabadán, C., Pardo-Pascual, J.E., Almonacid-Caballer, J., Rodilla, M., 2019. Detecting problematic beach widths for the recreational function along the Gulf of Valencia (Spain) from Landsat 8 subpixel shorelines. Appl. Geogr. 110, 102047. doi:10.1016/j.apgeog.2019.102047
Semeoshenkova, V., Newton, A., 2015. Overview of erosion and beach quality issues in three Southern European countries: Portugal, Spain and Italy. Ocean Coast. Manag. 118, 12–21. doi:10.1016/j.ocecoaman.2015.08.013
- explain the improvements of your study and its relevance in the international panorama
- you reported 4 objectives that, for their importance, could be considered as 4 individual research studies. Therefore, you must better describe their interrelations.
line 57: ICZM is the correct acronym. Bear in mind to explain every acronym before its use
line 67-68: avoid repetitions
line 68: The fundamentals of sedimentary transport have been studied extensively [19]. and then? please explain this concept
lines 70-76: you reported a series of literature citations without a clear logical direction. The same in part of your introduction that seems quite fragmented.
line 87: The resolution of the images has increased markedly, but it is not suitable for measuring medium to small coastal changes [32]. Here you should explain what you mean by medium/small changes. Changes or resolutions?
line 92-94: this sentence makes no sense
line 105-106: so here you should advance something about the erosion rates in past and recent decades. Moreover, you wrote about “problems”..which kind of problems?
line 105 to 120: references are completely lost
line 119-121: these affirmations are obviously true, but they are completely untied from the rest of the text
figure 1 and 2: include a map scale
which methodology did you apply for obtaining the textural parameters?
I suggest including for each type of survey, the monitoring/observing period.
In material and methods, you must insert the description about how you had marge the amount of data collected
lines 297-324: why did you duplicate here the study area? please avoid this part
fig 4: completely useless
mean scalar regime and significant waves…must be explained in-depth..
Fig.8: differences are not visible; therefore, you should draw the main geomorphological lines
Unfortunately, this study collected a lot of beach data without a clear methodological approach. In fact, each method step suffers from limitations and the whole manuscript reflects this tendency. Moreover, the authors had the ambition of integrating these steps for a comprehensive approach that。
Author Response
Reply to Reviewer 2
Introduction: I recommend rewriting the whole introduction considering that you should explain:
- the reasons behind this study in view of the literature gaps;
- if this study is an integrated approach of both physical, administrative, biological parameters, you should compare it with the most faculty studies on this field (there are dozens of studies about coastal management indicators and integrated approaches; see for example:
Cabezas-Rabadán, C., Pardo-Pascual, J.E., Almonacid-Caballer, J., Rodilla, M., 2019. Detecting problematic beach widths for the recreational function along the Gulf of Valencia (Spain) from Landsat 8 subpixel shorelines. Appl. Geogr. 110, 102047. doi:10.1016/j.apgeog.2019.102047
Semeoshenkova, V., Newton, A., 2015. Overview of erosion and beach quality issues in three Southern European countries: Portugal, Spain and Italy. Ocean Coast. Manag. 118, 12–21. doi:10.1016/j.ocecoaman.2015.08.013
- explain the improvements of your study and its relevance in the international panorama
- you reported 4 objectives that, for their importance, could be considered as 4 individual research studies. Therefore, you must better describe their interrelations.
The manuscript has been revised following your comments and suggestions. We appreciate the very detailed comments and suggestions that you have been noted. We believe that we have modified the original manuscript in a very important way, describing details of the study, reorganizing some sections and correcting deficiencies that you had been pointed out.
line 57: ICZM is the correct acronym. Bear in mind to explain every acronym before its use
Acronym has been explained before use. We have also changed it. The correct acronym to refer to the integrated coastal zone management is ICZM. We had made a mistake. Corrected
line 67-68: avoid repetitions
We have removed the repetition and rewritten
line 68: The fundamentals of sedimentary transport have been studied extensively [19]. and then? please explain this concept
We have rewritten this section. Additional information has been added.
lines 70-76: you reported a series of literature citations without a clear logical direction. The same in part of your introduction that seems quite fragmented.
References have been reordered to give consistency to the text. The objective is to refer to the problems of scale, the different fields and methods in which research work is applied in coastal areas.
line 87: The resolution of the images has increased markedly, but it is not suitable for measuring medium to small coastal changes [32]. Here you should explain what you mean by medium/small changes. Changes or resolutions?
This sentence has been rewritten. “The resolution of the images has increased markedly. But, although the resolution has increased significantly, it is still not suitable for measuring small-scale dimensional changes.”
line 92-94: this sentence makes no sense
It has been corrected
line 105-106: so here you should advance something about the erosion rates in past and recent decades. Moreover, you wrote about “problems”..which kind of problems?
Paragraph have been reorganized. Some additional explanations have been added
line 105 to 120: references are completely lost
Corrected
line 119-121: these affirmations are obviously true, but they are completely untied from the rest of the text
Section 2 has been deeply reorganized. Last paragraph has been reorganized. Some additional explanations have been added
figure 1 and 2: include a map scale
Done
which methodology did you apply for obtaining the textural parameters?
I suggest including for each type of survey, the monitoring/observing period.
It has been incorporated
In material and methods, you must insert the description about how you had marge the amount of data collected
This section has been also reorganized. Additional explanations have been added
lines 297-324: why did you duplicate here the study area? please avoid this part
Corrected
fig 4: completely useless
You were absolutely right. That figure has been removed. The data that figure showed have been incorporated into a table that allows you to see all the necessary details. See the new Table 2. Granulometric analysis, D50 size, % and type of sediments in Malavarrosa beach September 2021
mean scalar regime and significant waves…must be explained in-depth..
Your indications have been attended. Additional explanations have been added
Fig.8: differences are not visible; therefore, you should draw the main geomorphological lines
They have been incorporated. Additional explanations have been added
Thank you again for your comments and suggestions.

Reviewer 3 Report
This manuscript is is not really focused. On one hand, the authors describe some new methodology or dataset that is result of combining different datasets UAVs and remote sensing, and then they describe the situation of the Spanish Mediterranean coasts. In the end, there is no focus on this new approach, the discussion and conclusions are rather general.
The m/s is simply a combination of images collected by a drone and
historical data about granulometry and shoreline evolution. There is no
in-depth study or analysis. The English is poor too and some figures
don't contribute substantially to the m/s.
I think the authors are in conditions of collecting more data ana carry
out an analysis, whether comparative or temporal, to refocus this work
on some particular beaches.
Author Response
Reply to Reviewer 3
The manuscript has been revised following your comments and indications. We appreciate your comments and suggestions that you have been noted. We believe that we have modified the original manuscript in a very important way, describing details of the study, reorganizing some sections and correcting deficiencies that you had been pointed out.
Your indications have been attended. According to your suggestion we have reorganized some parts. We understand and hope that we have added clarity to the manuscript.
We have changed the acronym CAC and we have used CCA. We have removed some repetition and rewritten sections and paragraphs. We have corrected the acronym to refer to the integrated coastal zone management, ICZM.
Some paragraphs have been reorganized. Some of them have been removed. Some additional explanations have been added.
The sentences concerning “Zero line” has been eliminated. Zero line definition is established in section 3.2: “The study area does not have a significant tide. Small time scales sea level changes are mainly due to wind set-up, storm surges and wave conditions. The swash zone forms the land-ocean boundary at the landward edge of the surf zone, where waves runup and rundown the beach face. The ZL is defined by runup and rundown interpolation in the swash zone.”
We have explained, in more detail, where samples simples have been collected, determination of carbonates and minerology, the various methods of data collection and the transport rate calculation. the data used and the calculation performed are discussed in detailed.
Explanations detailing what was requested have been incorporated. Figures have been added. This section has been extensively reorganized following your comments and suggestions.
Figure 4 has been removed. The data that figure showed have been incorporated into a table that allows you to see all the necessary details. See the new Table 2. Granulometric analysis, D50 size, % and type of sediments in Malavarrosa beach September 2021
New explanations have been added in figure captions. We have added a new scheme in the figure 8 showing the location of the areas. Scale has been added. And we have made figures smaller. We have also explained that images are rotated to show the nord facing upwards.
We have rewritten some sentences in the results section and incorporated additional results. the influence of this amplitude of sea level changes in the width of the beaches is not relevant. The slope of the swash zone has an average slope of 34%. Consequently, variations in sea level of 12 centimeters imply maximum width changes of 35 centimeters.
Discussion section has been also rewritten. ICZM is a key element for the sustainable development of the coastal areas. One of the consequences of climate change is the increase in global coastal erosion and the loss of valuable coastal areas. A large number of development activities like construction of ports, urbanization, massive tourism, overexploitation of aquifers and the presence of dams, is increasing the vulnerability of coastal areas. Due to extraordinary storms on the coast, there is a great urgency for ICZM. Action must be addressed with a robust system based on coastal monitoring and the evaluation of results. Recent trends in coastal erosion mitigation is addressing towards soft and innovative methods, since hard methods have important impacts as high cost and down-drift erosion.
Thank you again for your comments and suggestions.

Round 2
Reviewer 3 Report
The manuscript looks better and some redundant information and figures were removed.
The quality of some figures still need some improvement.
Considering the importance of UAVs in this work, I recommend to modify the title to highlight this idea, the fact that you are combining the use of UAVs and maritime and sediment information.

Author Response
Reply to Reviewer 3 (Round 2)
“Comments and Suggestions for Authors:
The manuscript looks better and some redundant information and figures were removed.
The quality of some figures still need some improvement.
Considering the importance of UAVs in this work, I recommend to modify the title to highlight this idea, the fact that you are combining the use of UAVs and maritime and sediment information.”
The manuscript has been revised following your new comments and indications. We appreciate your comments and suggestions that you have noted. We have reorganized some figures and corrected deficiencies that you had pointed out. Your indications have been attended through the following actions:
- We have included in the title of the article. Final title we propose is: “Coastal monitoring using Unmanned Aerial Vehicles (UAVs) for the management of the Spanish Mediterranean coast: the case of Almenara–Sagunto”
- We have defined UAV in the Abstract
- In the Figure 1 we have increased size of the grid and North.
- We have corrected erratum in line 217.
- In the Figure 6 we have moved caption to right position. Also we have showed units in the graph axis
- Same in the Figure 7.
- In the Figure 10 we have increased size and corrected the subtitle.
Finally we have also decided to improve English language by using Editing Service MDPI.
Thank you again for your comments and suggestions.
